# SLOG: A Structural Generalization Benchmark for Semantic Parsing

**Bingzhi Li**[*,†]    **Lucia Donatelli**[λ]    **Alexander Koller**[℘]
**Tal Linzen**[μ]    **Yuekun Yao**[℘]    **Najoung Kim**[*,Δ]

[†]Université Paris Cité    [λ]Vrije Universiteit Amsterdam    [℘]Saarland University
[μ]New York University    [Δ]Boston University
bingzhi.li@yahoo.com, najoung@bu.edu

## Abstract

The goal of compositional generalization benchmarks is to evaluate how well models generalize to new complex linguistic expressions. Existing benchmarks often focus on *lexical generalization*, the interpretation of novel lexical items in syntactic structures familiar from training. *Structural generalization* tasks, where a model needs to interpret syntactic structures that are themselves unfamiliar from training, are often underrepresented, resulting in overly optimistic perceptions of how well models can generalize. We introduce SLOG, a semantic parsing dataset that extends COGS (Kim and Linzen, 2020) with 17 structural generalization cases. In our experiments, the generalization accuracy of Transformer models, including pretrained ones, only reaches 40.6%, while a structure-aware parser only achieves 70.8%. These results are far from the near-perfect accuracy existing models achieve on COGS, demonstrating the role of SLOG in foregrounding the large discrepancy between models' lexical and structural generalization capacities.

## 1 Introduction

Compositional generalization benchmarks that test the ability to understand novel utterances based on composition of known parts (Montague, 1974; Partee, 1984; Fodor and Pylyshyn, 1988) have emerged as a useful tool for model evaluation in semantic parsing. COGS (Kim and Linzen, 2020) in particular has become a widely-used benchmark, as it is designed to expose a generalization gap between training and testing data that many recent semantic parsers still struggle with.

COGS distinguishes two distinct types of generalization challenges: *lexical generalization* tests a model's ability to interpret novel combinations of known lexical items and known linguistic structures (Figure 1a), whereas *structural generalization*

---

*This work was conducted during Bingzhi Li's visit to NYU. The middle authors are listed in alphabetical order.

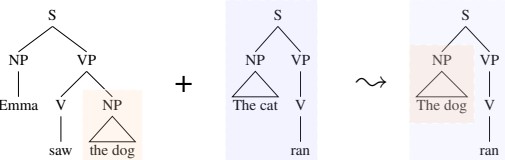

(a) Lexical generalization: object → subject (COGS)

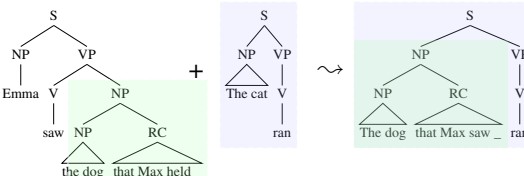

(b) Structural generalization: RC object→RC subject (SLOG)

Figure 1: Examples of lexical generalization in COGS (a), and structural generalization in SLOG (b). The SLOG task requires mapping the generalization examples to their logical forms; the corresponding logical forms are shown in Table 1.

tests the ability to combine known structures into a novel structure (Figure 1b). Importantly, most of the generalization types in COGS are lexical generalization (18 out of 21 generalization types, 86% of the dataset). As lexical generalization is arguably easier than structural generalization (e.g., solvable by simple slot-filling), this imbalance may lead to overall performance numbers that are overly optimistic with regard to a model's capacity to generalize compositionally (Yao and Koller, 2022).

To facilitate a more comprehensive evaluation of structural generalization, we introduce SLOG, a **S**tructural **LO**ng-distance dependencies **G**eneralization benchmark. SLOG extends COGS to include 17 cases of structural generalization in total (14 new cases and 3 existing cases from COGS) (§2). The novel generalizations we introduce target two key structural features of human language (§3): recursion and filler-gap dependencies.

We use SLOG to evaluate a sequence-to-sequence (seq2seq) Transformer model trained

| | Training | Generalization |
|---|---|---|
| COGS | Emma saw **the dog**. $\rightsquigarrow$ 
 $\star$dog$(x_3)$; see.agent$(x_1,$Emma$)$ $\wedge$ see.theme$(x_1, x_3)$ 
 The cat **ran**. $\rightsquigarrow$ $\star$cat$(x_1)$; run.agent$(x_2, x_1)$ | **The dog** ran. $\rightsquigarrow$ 
 $\star$dog$(x_1)$; run.agent$(x_2, x_1)$ |
| SLOG | Emma saw **the dog that Max held**. $\rightsquigarrow$ 
 $\star$dog$(x_3)$; see.agent$(x_1,$Emma$)$ $\wedge$ see.theme$(x_1, x_3)$ $\wedge$ dog.nmod$(x_3, x_6)$ $\wedge$ hold.agent$(x_6,$Max$)$ $\wedge$ hold.theme$(x_6, x_3)$ 
 The cat **ran**. $\rightsquigarrow$ $\star$cat$(x_1)$; run.agent$(x_2, x_1)$ | **The dog that Max saw** ran. $\rightsquigarrow$ 
 $\star$dog$(x_1)$; dog.nmod$(x_1, x_4)$ $\wedge$ see.agent$(x_4,$Max$)$ $\wedge$ see.theme$(x_4, x_1)$ $\wedge$ run.agent$(x_5, x_1)$ |

Table 1: Examples of lexical generalization in COGS and structural generalization in SLOG with their corresponding COGS logical form (LF) representation. The task requires mapping ($\rightsquigarrow$) the English sentences to their LFs.

from scratch (Vaswani et al., 2017), two pretrained Transformers (T5-base; Raffel et al. 2020 and LLaMA; Touvron et al. 2023), and a structure-aware[1] model (AM-Parser; Weißenhorn et al. 2022). In comparison to their overall performance on COGS, all models exhibit considerably lower performance on SLOG (§5). An error analysis reveals that the structure-aware AM-Parser generalizes well on the existing structural generalization cases in COGS but struggles with the gap constructions introduced in SLOG due to inherent structural limitations, which we discuss in §5.3. Transformers tend to erroneously repeat frequent meaning representation subsequences observed during training. Even with pretraining, they struggle with unseen long-distance dependencies, which we attribute to their bias towards shorter predicate-argument dependencies. Overall, the discrepancy in performance between SLOG and COGS demonstrates the utility of SLOG in exposing the overall limitations of current semantic parsing models shown to achieve high performance on existing generalization benchmarks, as well as highlighting the different weaknesses of these models.

## 2 The SLOG Benchmark

SLOG follows the semantic parsing format used in COGS, where the task is to translate English expressions into logic-based meaning representations. As in COGS, there is a systematic gap between the training set and the generalization set: there are constructions in the generalization set that are not included in the training set, but pieces of constructions included in the training set can be recombined to arrive at their correct meanings. For example, as illustrated in Table 1, noun phrases that appear

only in object position during training must be reinterpreted in subject position during generalization.

SLOG[2] is generated using manually specified rules (§3), adopting the same meaning representation as COGS. The COGS logical form (LF), derived from Reddy et al. (2017), uses indexed constants to represent entities or events. For example, in the first example of Table 1, $x_3$ denotes an entity that is both a dog and the theme of a seeing event, while $x_1$ denotes the seeing event. The constant names are determined by the linear position of the phrasal head in the input sentence.

SLOG contains 17 structural generalization cases grouped into four categories. These generalization cases are primarily motivated by frequency asymmetries in natural language, where simpler structures are more common than complex ones; in other words, SLOG assesses whether NLP models can extrapolate from frequent patterns to rare ones.

We describe the four categories below; see Table 2 for the full list of generalization cases.

### 2.1 Novel Recursion Depth

Recursion allows smaller phrases to be combined to create larger phrases. This combination process can be repeated an unbounded number of times. COGS tests a model's ability to apply recursion in two cases: sentential complements (tail CP recursion)[3] and nominal prepositional phrase modifiers (tail PP recursion). For both cases, the training set contains recursive depths of 0–2, where 0 indicates the absence of any PP or CP, and the generalization set contains the strictly greater depths 3–12.

By contrast, the SLOG training set includes recursion of depths 0–2 and 4, and the generalization

---

[1]In this paper, 'structure-aware' refers specifically to models that incorporate explicit representations of linguistic structure.

[2]The generation code and SLOG dataset are available at https://github.com/bingzhilee/SLOG.

[3]Nested clauses with right-branch embeddings like *[Max knows that [Mary knows [that Emma cooks]$_{CP}$]$_{CP}$]$_{CP}$*

| Generalization cases | Training | Generalization |
|---|---|---|
| | §2.1 Novel Recursion Depth | |
| *Deeper depth generalization* | | |
| ✓ Prepositional phrases (PP) max depth 4 → depth 5-12 | Ava saw the ball **in** the bottle **on** the table. | Ava saw the cat **in** the box **on** the mat **on** the bed **on** the floor **in** the room. |
| ✓ Tail CP recursion max depth 4 → depth 5-12 | Ava believed **that** Emma said **that** a fish froze. | Ava said **that** Emma liked **that** Max believed **that** Noah found **that** Liam saw **that** the cat slept. |
| Center embedding max depth 4 → depth 5-12 | Eva saw the cat **that** the horse **that** the dog liked chased. | Ava held the dress **that** a store **that** a girl **that** a boy **that** a cat **that** a man drew saw loved knew sold. |
| *Shallower depth generalization* | | |
| Prepositional phrases max depth 4 → depth 3 | Emma saw the ball **in** the bottle **on** the table **on** the floor **in** the office. | Ava saw the cat **on** the mat **on** the floor **in** the office. |
| Tail CP recursion max depth 4 → depth 3 | Ava believed **that** Emma said **that** Max found **that** a cat saw **that** a fish froze. | Ava said **that** Emma liked **that** Max believed **that** the cat slept. |
| Center embedding max depth 4 → depth 3 | Eva saw the cat **that** the horse **that** the dog **that** the man **that** the girl loved found liked chased. | Emma bought the dress **that** the store **that** the woman **that** Mike knew liked sold. |
| | §2.2 Novel Combination of Modified Phrases and Grammatical Roles | |
| PP in direct object NPs | | |
| ✓ → PP in subject NPs | Noah ate **the cake on the plate**. | **The cake on the table** burned. |
| → PP in indirect object NPs | Noah ate **the cake on the plate**. | Max gave a fish to **a cat on a table**. |
| RC in direct object NPs | | |
| → RC in subject NPs | Noah saw **the cat that froze**. | **The cat that froze** smiled. |
| → RC in indirect object NPs | Noah saw **the cat that froze**. | Max gave a fish to **a cat that ran**. |
| | §2.3 Novel Gap Positions | |
| Subject, direct object-extracted RC → Indirect object-extracted RC | Noah saw the cat that gave a fish to Liam. ⊕ Noah saw the cat that Liam liked _. | Noah saw the cat that Emma gave a cake to _ . |
| Subject, direct object *wh*-questions → Indirect object *wh*-questions | Who saw the cat? ⊕ What did Emma see _? | Who did Noah give the cake to _? |
| | §2.4 Novel *wh*-questions | |
| Subject, object *wh*-Q of simple transitives | | |
| → Active subject *wh*-questions | **Who saw** the cat? ⊕ Emma **wanted** to sleep. | **Who wanted** to sleep ? |
| → Passive subject *wh*-questions | **Who** did Emma see _? ⊕ The boy **was found** by Emma. | **Who was helped** by Emma? |
| → Direct object *wh*-questions with ditransitive verbs | **What** did Emma see _? ⊕ Emma **gave** a fish **to** the cat. | **What** did Emma **give** _ to the cat? |
| → *Wh*-questions with modified NPs | What did **the cat** see _? ⊕ the cat **on the mat** | What did **the cat on a table** see _? |
| → *Wh*-questions long movement | **What** did the cat **see** _? ⊕ Emma **said that** the cat saw a fish. | **What** did Emma **say that** the cat **found** _? |

Table 2: A full list of SLOG generalization cases. Each sentence in the table corresponds to a (sentence, logical form) pair, as illustrated in Table 1. ⊕ denotes the composition of two observed structures, which allows to interpret the target novel structure. Some cases cover multiple constructions: e.g., all ditransitive verbs include both double-object and prepositional constructions. The three cases marked with '✓' are structural generalization cases already present in the COGS dataset.

set contains both the intermediate depth 3 and the greater depths 5–12. Including both shallower and deeper embeddings in the generalization set allows us to determine if any difficulty in generalizing to an unseen embedding depth is a consequence of the model's more general difficulty in processing longer sequences than observed in training (Lake and Baroni, 2018; Herzig et al., 2021; Anil et al., 2022) rather than a more specific issue with applying recursion to generate novel constructions.

In addition to this new depth split, SLOG introduces a new recursion construction. COGS involves only tail recursion, which features recursive PPs and CPs with right-branch embeddings. SLOG extends this with center embedding, where a phrase is embedded in the middle of another of the same type, leaving elements on both sides of the embedded component and producing well-parenthesized long-distance dependencies, as denoted by the subscript numbers:

(1) Eva saw the mouse [that the $cat_1$ [ that the $dog_2$ $chased_2$ ] $held_1$ ].

At the same recursion depths, the average LF length increases from PP recursion to tail CP recursion to center embedding.

In natural language, recursion depth is rarely greater than five, and center embedding is generally limited to two levels (Karlsson, 2007, 2010). By contrast, SLOG includes recursion up to depth 12. While this may surpass human processing abilities for reasons presumed to be linked to memory constraints (Gibson and Thomas, 1999; Karlsson, 2007), deeper embedding depth remains grammatical, echoing Chomsky's competence versus performance distinction. Importantly, we also note that our goal with SLOG is to evaluate the linguistic competence of NLP models, whose goal is not to simulate human performance limitations.

## 2.2 Novel Combination of Modified Phrases and Grammatical Roles

SLOG also tests the capacity to interpret complex noun phrases (NPs) in new positions. In addition to PP modifiers included in COGS, we introduce relative clause modifiers.

### 2.2.1 Prepositional Phrase Modifiers

In COGS, NPs modified by PPs are seen only as direct objects (2), and need to be interpreted as subjects during generalization (3). SLOG adds generalization cases targeting indirect object modification (4).

(2) Noah saw [a cat on the table]$_{dobj}$.

(3) [The **cat** on the mat]$_{subj}$ **ran**.

(4) Emma **gave** [a cat on the mat]$_{iobj}$ a **fish**.

We expect sub-cases of indirect object modification to pose challenges of varying difficulty, depending on the length of the predicate-argument dependency. In particular, generalization to indirect object modification in active oblique datives (4) introduces a dependency between the verb *gave* and the direct object *a fish* across the non-argument NP *the mat*.[4] In contrast, sub-cases like (5a) and (5b), where the non-argument NP occurs at the end of the sentence, do not include a dependency across an intervening NP; we therefore expect them to be relatively easier.

(5) a. Emma gave a fish to [a cat on the mat]$_{iobj}$.

b. A fish was given to [a cat on the mat]$_{iobj}$.

SLOG's training set additionally includes standalone PP-modified NPs (e.g., the NP *the cat on the table* on its own[5]) to prevent modifiers from being associated with only a particular range of token indices, as pointed out by Wu et al. (2023).[6] Such standalone NPs, which are common in child-directed speech (Wells and Bridges, 1981; Cameron-Faulkner et al., 2003) but were not a part of COGS, serve as a signal that the distribution of PP-modified NPs is not restricted to the object position.

### 2.2.2 Relative Clause Modifiers

Similar to PP modifiers, NPs with relative clause (RC) modifiers, as in (6), can occupy any position that an unmodified NP can fill. We expect RC modifiers to pose a greater challenge compared to PP modifiers, as they involve *gap constructions*, in which a phrase needs to be interpreted in a position other than its canonical position in a declarative clause. We refer to this as *extraction* (Sag, 2010), and we mark gap positions with an underscore. In (6), *the dog* should be interpreted as if it occupies the gap position as the direct object of *held*; in the logical form, this is represented by the fact that $x_3$ is filling both see.theme and hold.theme.

---

[4]This observation also holds true for the generalization to subject modification shown in (3).

[5]the cat on the table $\leadsto$ *cat$(x_1)$; *table$(x_4)$; cat.nmod.on$(x_1, x_4)$

[6]PPs in COGS were restricted to the object position, so models never observed the association of modifiers with linearly-earlier indices, which makes it difficult to isolate this effect from structural generalization.

(6) Emma saw the dog that Max held __.

$\leadsto$ `*dog(`$x_3$`);` `see.agent(`$x_1$`, Emma)` $\wedge$ **see.theme**$(x_1,\ x_3)$ $\wedge$ `dog.nmod(`$x_3, x_6$`)` $\wedge$ `hold.agent(`$x_6$`,` `Max)` $\wedge$ **hold.theme**$(x_6,\ x_3)$

Similar to the case of the PP modifiers (§2.2.1), the training set contains direct object NPs modified by RCs as well as standalone RC-modified NPs, as in (7). The generalization set contains RC modifiers for subject NPs, as in (8a), and indirect object NPs, as in (8b):

(7) TRAINING

   a. Liam saw [the cat that Emma held __]$_{dobj}$ .

   b. the cat that Liam fed __

(8) GENERALIZATION

   a. [The cat that Emma found __]$_{subj}$ smiled.

   b. Liam gave [a cat that Emma held __]$_{iobj}$ a fish.

## 2.3 Novel Gap Positions

The SLOG training set contains both subject and direct object extraction in RCs (9); these are the most frequent extraction positions in both written and spoken English corpora (Roland et al., 2007). The generalization set includes extraction of indirect objects (10), a less frequent construction.

(9) TRAINING

   a. Liam saw the boy that ate a cake.

   b. Liam saw the boy that Emma loved __

(10) GENERALIZATION

   a. Liam saw the boy that Emma gave a cake to __ .

SLOG also tests for the interpretation of novel gap positions in *wh*-questions. As with RCs, the training set includes questions with either subject or direct object extraction (11), and the generalization set contains questions with indirect object extraction (12).

(11) TRAINING

   a. Who did Emma love __?

   b. Who ate a cake?

(12) GENERALIZATION

   a. Who did Emma give a cake to __?.

In a *wh*-question (11a), a *wh*-filler (*who*) in the initial position of the clause is interpreted as if it occupied the gap (again indicated with an underscore) in the direct object position of *love*.

## 2.4 Novel *Wh*-questions

Next, we evaluate generalization to extraction cases that involve familiar gap positions—subject and direct object—paired with verb types that have never been observed in *wh*-questions during training. For this case, the training set contains *wh*-questions with simple transitive verbs (13) and declarative sentences with various verb types: transitive, intransitive and ditransitive. The generalization set includes five novel types of *wh*-questions that have not been observed during training, though their declarative counterparts have.

The novel *wh*-questions have varying distance between the *wh*-filler and the gap. Subject *wh*-questions, which maintain the same word order as their declarative counterparts, exhibit no gap (14a, 14b). Questions about direct objects of ditransitive verbs (14c), as well as questions with NPs modified by either a PP or an RC (14d),[7] have moderately long filler-gap distances. The filler-gap distance is longest for object extraction out of embedded clauses (14e).

(13) TRAINING

(The training set also includes the declarative counterparts of (14).)

   a. Who **saw** a cat ?

   b. What did Emma **see** __?

(14) GENERALIZATION

   a. Who froze ?

   b. What was frozen ?

   c. What did the boy give __ to Liam?

   d. What did Max give a cat that slept __?

   e. What did a boy say that Max believed that the cat saw __?

## 3 Dataset Generation

**Grammar** SLOG is generated from a probabilistic Synchronous Context-Free Grammar (SCFG) implemented in Alto (Gontrum et al., 2017). This grammar simultaneously generates the English expressions and their corresponding meaning representations (see Appendix B for more details).

**Training and generalization sets** We follow a similar sampling procedure to COGS. A total of

---

[7]*Wh*-questions with PP- or RC-modified NPs include various constructions where modifiers appear in subjects, direct objects, or indirect objects, exhibiting an average filler-gap distance similar to ditransitive verb *wh*-questions.

10,607 sentences are sampled from the probabilistic SCFG and then split into training, in-domain validation and in-domain test sets with an 8:1:1 ratio. The splits are then merged with the corresponding COGS splits. We then add 100 standalone PP-modified NPs and 100 standalone RC-modified NPs to the training set, as discussed in Section 2.2.

We also include what we refer to as primitive exposure examples for each ditransitive verb and verb accepting CP arguments,[8] totaling 40 primitives. These are standalone verb lexical meanings, such as, *hope* $\rightsquigarrow \lambda a. \lambda b. \lambda e.\texttt{hope.agent(e,b)} \land \texttt{hope.ccomp(e,a)}$. This results in a final training set of 32,755 examples and 4,046 examples in both validation and in-distribution test sets.

For the generalization set, we use separate grammars for each generalization case. We sample 1000 examples from each of the 17 cases, yielding a total of 17,000 examples. For the training set and the generalization set, the maximum lengths of the input English sentences are 28 and 61 tokens, respectively. The maximum lengths of the corresponding output logic forms are 229 and 599 tokens. See Appendix B for more details.

## 4 Experimental Setup

**Models** We evaluate the performance of seq2seq, autoregressive, and structure-aware models on SLOG. The seq2seq models we evaluate are a Transformer we train on SLOG from scratch (*vanilla Transformer* henceforth; Vaswani et al. 2017), and a finetuned pretrained Transformer (T5; Raffel et al. 2020) that has demonstrated strong performance on multiple compositional generalization tasks (Herzig et al., 2021). The autoregressive Transformer model we evaluate is LLaMa (Touvron et al., 2023). Finally, the structure-aware model we evaluate is the AM-Parser (Groschwitz et al., 2018), which achieves near-perfect accuracy on COGS (Weißenhorn et al., 2022). Previous work has shown that structure-aware models perform well on compositional generalization tasks, specifically those involving structural generalization (Yao and Koller, 2022). Following Weißenhorn et al. (2022), we first have the AM-Parser predict an intermediate dependency tree, and then convert it to a graph-based representation of the SLOG logical form. We use the A* AM-parser from Lindemann

---

[8]Primitive examples of these two verb types let us incorporate their infinitive forms, used in *wh*-questions, into SLOG's vocabulary.

et al. (2020) for our experiments, as it yields the best overall results compared to alternative versions of AM-parser, such as the one in Groschwitz et al. (2018).[9] We run each experiment with five different random seeds. See Appendix A for more details.

**Evaluation metric** Most studies report exact match accuracy on COGS. This metric has two limitations that may lead to an underestimation of a model's generalization capacity. First, because the COGS LF is conjunctive, reorderings of the conjuncts are semantically equivalent; yet, under exact match accuracy, only a single order is considered correct. Second, the COGS LF uses Skolem constants with a naming scheme tied to the linear indices of phrasal heads in the input. While a commitment to a systematic naming scheme is necessary for consistent evaluation, different naming schemes up to the renaming of the constants in the gold LF yield equivalent LFs (e.g., (15a) vs. (15b)). Such LFs would be considered incorrect under exact match.

To incorporate semantic equivalence up to conjunct reordering and constant renaming, at evaluation time, we alphabetically sort the conjuncts of the gold LFs, and subsequently index variables based on their appearance order in the sorted LFs. The same modifications are applied to the model outputs. This process results in the reformatted output as shown in (16); applying these modifications to (15a) and (15b) yields the same outcome. Then, computing exact match on these postprocessed LFs captures the targeted semantic equivalence.

(15) Gold LF and model-predicted LF for *What did the baby eat?*:

  a. Gold: $\texttt{eat.theme}(x_4, ?) \land \texttt{eat.agent}(x_4, x_3) \land \texttt{baby}(x_3)$

  b. Out: $\texttt{eat.agent}(x_3, x_6) \land \texttt{eat.theme}(x_3, ?) \land \texttt{baby}(x_6)$

(16) Reordered and reindexed version:

  a. $\texttt{baby}(y_2) \land \texttt{eat.agent}(y_1, y_2) \land \texttt{eat.theme}(y_1, ?)$

This reformatted exact-match metric is used for all results reported in the main text; see Appendix C.1 and Table 5 for more details.

## 5 Results

Overall, seq2seq Transformers, both trained from scratch and pretrained, display low accuracy on

---

[9]For a detailed discussion, please refer to Appendix D.

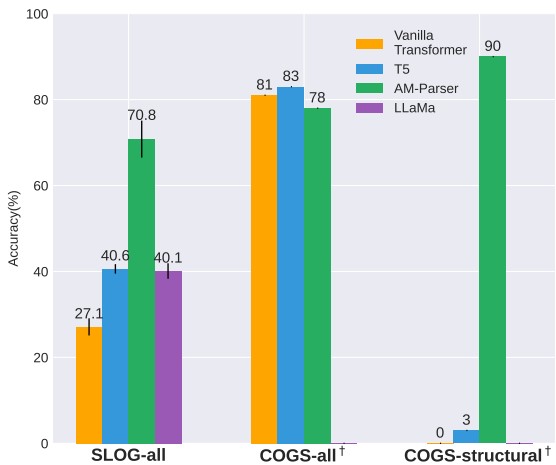

Figure 2: Accuracy on SLOG, with error bars indicating variations across five runs. We also show the best published results on COGS (indicated with [†]), as reported in Yao and Koller (2022).

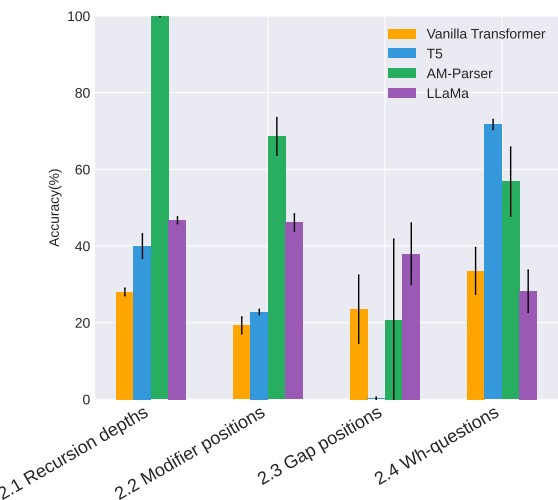

Figure 3: Aggregate accuracy on SLOG by generalization category, with error bars denoting the standard deviation across generalization cases within each category over five model runs.

SLOG (Figure 2), in line with earlier studies on structural generalization in seq2seq models (Yao and Koller, 2022). This is also the case for the more recent autoregressive Transformer LLaMa, whose performance is similar to that of T5. As Figure 2 shows, high accuracy on the full COGS dataset, where 86% of the generalization cases are lexical, can obscure low performance on structural generalization, highlighting the need for the expanded structural generalization tests included in SLOG.

SLOG additionally reveals weaknesses in the AM-Parser that COGS did not. While the AM-Parser achieves 90% accuracy on the structural generalization subset of COGS (Figure 2), it faces systematic difficulties with several generalization types introduced in SLOG (Figure 3).

Performance varied substantially across generalization categories (Figure 3); in particular, all models achieved near-perfect accuracy on *Active subject wh-questions* and *Shallower PP recursion*. These cases were the least structurally complex in their respective categories (§2.3 and §2.1).We highlight specific error types in the rest of this section; see Appendix C for full results and additional error analysis.

## 5.1 Unobserved Depth and Length Both Affect Depth Generalization

The maximum depth observed in training was four levels of embedding for all three recursive structures tested. All models achieve greater than 90% accuracy on unseen shallower PP recursion (three

levels of embedding). A considerably lower performance is observed for seq2seq models with shallower tail CP recursion (<61%); in particular, the Transformer trained from scratch consistently fails to generalize to shallower center embedding, with zero accuracy overall. Transformer models show systematically lower performance on deeper recursions (5-12 levels of embedding), whereas the structure-aware model is robust to depth variation.

| | Vanilla Transformer | T5 | LLaMa | AM parser |
|---|---|---|---|---|
| *At or below max training output length* | | | | |
| PP recursion | 29.3 | 37.0 | 46.0 | 100.0 |
| Tail CP recursion | 3.0 | 17.7 | 40.2 | 100.0 |
| Center embedding | 0.0 | 0.0 | 0.0 | 100.0 |
| *Beyond max training output length* | | | | |
| PP recursion | 0.0 | 0.0 | 0.0 | 100.0 |
| Tail CP recursion | 0.0 | 0.0 | 0.0 | 100.0 |
| Center embedding | 0.0 | 0.0 | 0.0 | 100.0 |

Table 3: Mean accuracy (%) on unseen deeper recursion cases, broken down by whether the expected output falls within or exceeds the range of training output lengths (maximum training output = 229 tokens).

We investigate the relation between length and depth generalization further by dividing the deeper depth generalization cases into examples that are shorter vs. longer than the maximum output length observed in training (229 output tokens). Results are shown in Table 3. All tested Transformer models are unable to generalize to examples longer than the maximum output length observed in training;

this result is consistent with the difficulty of length extrapolation observed in the literature (Hupkes et al., 2020; Anil et al., 2022). Length extrapolation does not capture the full story, however: the model's performance is limited even when the length of the generalization examples falls within the range of observed output lengths. This indicates that unobserved depth indeed plays a role in these models' poor generalization to deeper structures, in addition to known difficulties in length generalization.

## 5.2 Unobserved Long-distance Dependencies Make Generalization Difficult

Generalizing to subject modification (both PP and RC) is one of the most challenging cases. Seq2seq models achieve near-zero accuracy, even with the additional cue from the standalone modified NPs that modification can appear outside of object positions. This challenge echoes previous findings on COGS (Akyurek and Andreas, 2021; Zheng and Lapata, 2022; Yao and Koller, 2022). The remainder of this section focuses on the analysis of PP modification cases, but similar patterns are observed for RC modifiers, which we discuss in Appendix C.3.

Common error patterns across Transformer models reveal a bias towards shorter predicate-argument dependencies. For instance, in sentences like *A* **cat** *on the mat* **froze**, models often misinterpret the closer NP *the mat* as the subject.

A further breakdown of the modifier generalization performance by construction shows that examples involving longer predicate-argument dependency (i.e., there is an intervening non-argument NP between the predicate and the argument) tend to be more difficult for all models (Table 4). However, the Transformer-based models show a stronger bias towards linearly adjacent predicate-argument structures. Further analysis (Appendix C.2) shows that seq2seq models additionally fall prey to inference patterns akin to a modification rule "attach PPs to NPs in immediate post-verb position", which is compatible with the training data but leads to incorrect generalization.

## 5.3 Gap Generalizations Are Challenging for All Tested Models

For gap generalization cases, all models display low accuracy and considerable variability across different runs as shown in Figure 3. While Transformer models are biased towards more frequent subsequences of output LFs observed during train-

ing (see Appendix C.4), the structure-aware AM-Parser demonstrates different generalization difficulties.

The AM-Parser systematically fails on every instance of *wh*-questions involving long movement (e.g. *What did Ava say that the cat saw ___?*). This issue arises from its internal prediction of dependency trees, which represent how meaning representations are compositionally constructed. For these *wh*-questions, the required dependency trees are nonprojective since the edge from the embedded verb to the *wh*-pronoun crosses the matrix verb. However, the AM-Parser used in our study only supports projective dependency trees, leading to incorrect prediction of sentence structure.[10] This issue with projectivity can serve as a diagnostic for structural limitations of similar structure-aware parsers (Liu et al., 2022; Qiu et al., 2022a).

Furthermore, on the indirect and direct object *wh*-questions, the AM-Parser performs very unpredictably, with accuracies ranging from 0 to 80 depending on the random seed. This is because at the bottom of its compositional process, the AM-Parser predicts the lexical meaning for each token in the sentence (*supertag*). In these generalization types, the gold meaning representations in the test set require supertags that are infrequent in training. Thus, while the AM-Parser can compensate the distribution shift of the meaning representations as a whole, SLOG exposes its weakness to distribution shifts in the lexical supertags. A more detailed discussion is provided in Appendix D.

## 6 Related Work

Previous research has shown that recurrent neural network (RNN) architectures often struggle with learning complex long-range relations from simpler formal languages (Avcu et al., 2017; Mahalunkar and Kelleher, 2019). Our results on SLOG reveal that unseen long-distance predicate-argument dependencies pose considerable difficulty for Transformer-based models as well (§5.2). For filler-gap dependencies, prior work has centered on syntactic tasks involving *wh*-questions or relative clauses (Wilcox et al., 2018; Marvin and Linzen, 2018; Li et al., 2023; i.a.). These studies primarily use language modeling as the task and do not require mapping to semantic representations. SLOG incorporates both long-distance predicate-

---

[10]Alternative versions of the AM-Parser that can handle non-projective trees exist and are discussed in Appendix D.

| Generalization cases | Long pred-arg dependency? | Vanilla Transformer | T5 | LLaMa | AM parser |
|---|:---:|:---:|:---:|:---:|:---:|
| Sub-case: Passive indirect objects
**A fish was given** to [ a cat on the mat ]$_{iobj}$. | ✗ | 95.5 | 97.5 | 98.2 | 93.6 |
| Sub-case: Indirect object in PP datives
Emma **gave a fish** to [ a cat on the mat ]$_{iobj}$. | ✗ | 22.9 | 50.5 | 75.5 | 100.0 |
| Sub-case: Indirect object in double object datives
Emma **gave** [ a cat on the mat ]$_{iobj}$ **a fish**. | ✓ | 4.5 | 9.7 | 36.3 | 77.9 |
| Subject
[**A cat** on a mat]$_{subj}$ **ate** a fish. | ✓ | 0.0 | 0.8 | 28.9 | 57.6 |

Table 4: Performance of PP modification generalization broken down by construction. Bold orange words denote long predicate-argument dependencies, while bold black words indicate short ones.

argument and filler-gap dependencies within a semantic parsing setup.

Generalizing recursive patterns to deeper structures has been investigated in both artificial neural networks and humans using artificial languages (Christiansen and Chater, 1999; Lakretz et al., 2021; McCoy et al., 2021). Our findings underscore Transformer-based models' limitations with deeper recursive structures, corroborating the observations of Hupkes et al. (2020); Lakretz et al. (2021). In contrast, human studies have shown that they can learn and extrapolate center-embedding patterns to greater depth in artificial languages (Fitch and Hauser, 2004; McCoy et al., 2021).

Generalization cases in SLOG draw inspiration from the frequency gaps in natural language, where common patterns serve as a foundation for generalizing to rarer structures. This has connections to language acquisition in children, who have limited exposure to complex, less frequent structures, yet need to generalize to novel complex utterances by extrapolating from familiar linguistic elements (Perfors et al., 2011; Tomasello and Olguin, 1993; Atkinson et al., 2018). Human proficiency in such generalizations is attributed to inductive biases rooted in systematic compositional rules. However, the Transformer-based models we tested, despite excelling in lexical generalization scenarios, face challenges when presented with unfamiliar linguistic structures requiring such rule induction, hinting at potentially different or inadequate underlying mechanisms. More broadly, how the compositional generalization cases proposed in this work can be connected to human language acquisition is an interesting area of future study.

## 7 Conclusions

We introduce SLOG, a semantic parsing dataset that extends the COGS benchmark with a focus on structural generalization, which is often underrepresented in current benchmarks for compositional generalization. Using SLOG, we assess the structural generalization capacity of Transformer models (both pretrained and trained from scratch), as well as AM-Parser, a structure-aware model. While all models achieve good overall accuracy on COGS ($\geq$ 78%), their performance on SLOG is substantially lower, especially for Transformer models ($\leq$ 41%). Furthermore, even the structure-aware AM-Parser, which achieved strong performance on all structural generalization cases of COGS, performs poorly on several of the newly introduced generalization types in SLOG. Our error analysis shows that all Transformer models tested struggle with interpreting unseen long-distance dependencies and deeper recursive constructions than observed in training. On the other hand, the AM-Parser, despite its stronger overall performance (71%), displays categorical failures on gap generalization due to its inherent parser design limitations. Overall, SLOG exposes the limitations of a range of models that have previously been claimed to achieve good compositional generalization, and can serve as a useful analytic tool for guiding future improvements.

## Limitations

SLOG is a synthetic corpus and covers only a fraction of the diverse structures in English. Furthermore, previous research has demonstrated that the design of meaning representations (MR) can have a nontrivial effect on model performance in semantic parsing tasks (Guo et al., 2019; Herzig et al., 2021; Qiu et al., 2022b). For example, as noted by Wu et al. (2023), the variable indexing scheme

may introduce additional semantically irrelevant challenges when assessing structural generalization. SLOG's reformatted exact-match evaluation metric partially addresses this concern by taking into consideration several variations of MRs that are semantically equivalent, including MRs that are equivalent up to constant renaming. However, a more comprehensive study of the effect of artifacts from the formalism is left to future work.

There also exist challenges specific to the evaluation of pretrained models. That is, distributional shift between training and generalization sets intended by SLOG, such as withholding the constructions *PPs modifying subject NPs* from training, is difficult to strictly enforce when pretraining is involved (Kim et al., 2022). This potential violation of distributional control makes the interpretation of the obtained results difficult; we cannot disentangle whether generalization success in pretrained models derives from genuine compositional capabilities or simply exposure during pretraining to the target constructions meant to be withheld from the evaluated models. Still, corpus analyses such as Karlsson (2007) suggest that deep center embedding beyond three levels is very rare in naturally occurring data, so it is possible that very deep embedded structures are withheld as intended even from models exposed to large amounts of pretraining data. We hope the additional structural generalization cases that SLOG offers can also help with future work investigating the interaction between structures available in pretraining data and structural generalization.

## Acknowledgments

We thank Zhengxuan Wu, Christopher Manning, Christopher Potts and all members of the NYU Computation and Psycholinguistics Lab for helpful discussion. This work was supported in part through the NYU IT HPC resources, services, and staff expertise, and was funded by Labex EFL ANR-10-LABX-0083, the laboratory LLF of Université Paris Cité, the DFG through project KO 2916/2-2, and the National Science Foundation (NSF) under Grants No. BCS-204122, BCS-2114505 and IIS-2239862.

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

## A   Training details

**Hyperparameters**   The architecture of the Transformer trained from scratch is the same as in original COGS, which consists of 2 encoder and 2 decoder layers, 4 attention heads per layer, and a feedforward dimension of 512. We use the best combination of hyperparameters from Csordás et al. (2021) on COGS: a learning rate of 0.0001 with no label smoothing, warmup, or early stopping. Absolute positional embeddings with down scaling scheme (He et al., 2015; Csordás et al., 2021) is used due to stability issues observed with relative positional embeddings in recursive depth generalization cases, a similar phenomenon also noted in

Csordas and colleagues' experiments. Models are trained for 50k steps with a batch size of 128.

For the T5 experiments, we finetune T5-base[11] using a learning rate of 1.5e-5 and no label smoothing, warmup or early stopping. We finetune the model for 50k steps using a batch size of 2048.

For the LLaMA experiments, we finetune `llama-7b-hf`[12] with LoRA (Hu et al., 2021).[13] We set the learning rate to 3e-4, LoRA rank to 8, alpha to 32 and dropout to 0.1. We finetune the model for 5K steps with a batch size of 64, with 100 warmup steps and no label smoothing or early stopping. We apply LoRA to $W_q$ and $W_v$ weight matrices in the model.

All our experiments were run five times, using different random seeds. The final checkpoints from each run were used for evaluation on both in-domain test and out-of-domain generalization sets.

## B  Data generation

### B.1  Meaning representations

We use Alto (Gontrum et al., 2017) to implement a probabilistic Synchronous Context-Free Grammar (SCFG), which simultaneously generates pairs of English expressions and their corresponding meaning representations. Since SCFG cannot handle logical variables (Wong and Mooney, 2007), we use a variable-free representation proposed by Qiu et al. (2022a) (17a) as an intermediate representation during generation. The variable-free logical form (LF) can be deterministically postprocessed into the original COGS LF (17b) with additional information and specific constraints: (i) We rely on the word order in the input sentence to label the Skolem constants (i.e. variables); (ii) While the variable-free LF is unable to represent binding relations correctly as pointed out by Wu et al. (2023), an additional constraint that disallows duplicate nouns enables the intended binding relations to be identified unambiguously.

(17)  A cat slept. $\rightsquigarrow$

    a.  Variable-free LF:
       `sleep(agent=cat)`

    b.  COGS LF:
       `cat`$(x_1)$ `∧`
       `sleep.agent`$(x_2, x_1)$

[11] https://huggingface.co/t5-base
[12] https://huggingface.co/spaces/tloen/alpaca-lora
[13] https://github.com/tloen/alpaca-lora

(18)  A cat wanted to sleep. $\rightsquigarrow$

    a.  Variable-free LF:
       `want(agent=cat,`
       `xcomp=sleep(agent=cat))`

    b.  COGS LF:
       `cat`$(x_1)$ `∧`
       `want.agent`$(x_2, x_1)$ `∧`
       `want.xcomp`$(x_2, x_4)$ `∧`
       `sleep.agent`$(x_4, x_1)$

In the original COGS LF, entities or events specified by the predicates are represented by indexed constants (17b). In its variable-free counterpart (17a), `sleep` denotes the sleeping event, `cat` expresses the existence of a cat entity and fills the `agent` role of the sleeping event. In this way, each predicate in the LF has a set of arguments directly connected to their thematic roles without using variables.

Since the variable-free LF often results in a more compact LF, it has been adopted as the primary meaning representation in several prior work (Qiu et al., 2022b; Drozdov et al., 2022). We move away from this practice and keep the original COGS LF as the main meaning representation—as briefly mentioned above, the variable-free LF cannot represent binding relations accurately unless some external heuristic or constraint is introduced for disambiguation. For example, the variable-free LF in (18a) is ambiguous between the meaning of *A cat wanted to sleep* and *A cat wanted a (different) cat to sleep*, whereas the COGS LF in (18b) unambiguously represents the meaning of *A cat wanted to sleep*.

While we release the SLOG dataset in both LFs and report the results using the variable-free LF in Appendix E to enable comparison with existing work, we strongly recommend using the original COGS LF for evaluation on SLOG in future work.

### B.2  Grammar and sampling details

SLOG expands upon the COGS vocabulary, which consists of 503 nouns and 113 verbs, to additionally include *wh*-words (*who, what*) and *that* used as a relative pronoun. In SLOG, for the sake of simplicity, we only consider restrictive relative clauses introduced by *that* regardless of the animacy of the head NPs. For indirect object-extracted instances, we use the preposition stranding structure, such as *the boy that Emma give a cake to*, rather than *the boy to whom Emma gave a cake*.

The dataset includes the 30,000 examples from the initial COGS training set, and new examples that fall into one of the following categories:

- Relative clauses within object NPs, equal in number to instances with PP modifications

- Subject and object *wh*-questions matching the quantity of their corresponding declarative sentences

- An equal number of four-level-nesting recursion constructions as the depth-2 instances in initial COGS

- A primitive example for each ditransitive verbs and verbs accepting complement clause (CP) arguments

Finally, the SLOG sampling process excludes sentences with duplicate nouns (e.g. *Emma saw Emma.*), as mentioned in Section B.1.

**Semantic plausibility** Following COGS, our grammar implements simplified selectional restrictions, focusing mainly on animacy constraints. For instance, the subjects of unergative verbs are limited to animate entities, as in *the cat smiled*. As a result, our generated sentences may include semantically odd but syntactically well-formed sentences, such as non-edible object being the theme of *eat* or spatial incongruities like *a house in a bottle*. While these semantic limitations are unlikely to affect models trained from scratch, they may influence the performance of models that have been pretrained on naturalistic language data. It's important to note that our primary aim is to assess the extent to which models rely on compositional structural generalization to derive meaning. In line with the classic example "colorless green ideas sleep furiously" (Chomsky, 1957), which demonstrates that syntactic structure can be independent of semantic coherence, we argue that a model capable of compositional generalization should be able to map such sentences to an appropriate logical form as long as they are structurally well-formed.

**Structural disambiguation choice** In SLOG, mappings to logical forms are designed to be unambiguous, particularly for sentences that are inherently ambiguous due to prepositional phrase attachment ambiguity, such as *Ava saw the ball in the bottle on the mat*. This design choice, following COGS, is to use right-branching disambiguation for

all meaning representations. Consequently, SLOG ensures that PP modifiers are consistently interpreted as nested NP-attachments—*Ava saw [the ball [in the bottle [on the mat]]]*, although a VP-attachment might sometimes seem more intuitive depending on the context. This approach ensures that there exists an unambiguous target meaning representation for each expression in the dataset (and this is clearly signaled by the training data), rather than preserving the ambiguity which may complicate the evaluation protocol.

## C Full results and additional analyses

All models perform very well on the in-domain test set (accuracy over 99%). All experiments in this work were conducted on the out-of-domain generalization set, and we report the full results of the experiments discussed in Section 5 in Table 5.

### C.1 Effect of the reformatted exact-match metric

All models exhibit higher overall accuracy with the reformatted exact-match evaluation compared to the initial metric, notably pretrained models with an increase of over 10 percentage points (Table 5). This suggests that the initial exact-match metric may have underestimated model performance.

### C.2 PP Modifiers in unseen positions

As discussed in Section 5.2, generalization to PP modification involving unseen long predicate-argument dependencies is challenging for all evaluated models. Among such constructions, PP modification in the indirect object position (20a) is less challenging than subject position (19). A possible explanation is that the former has a closer surface resemblance to direct object modification—modifiers attach to an immediate post-verb NP. Indeed, we observe that a higher proportion of indirect object modifications are partially correct; models correctly predicted the PP-modified NP, but erred in the argument structure.

Table 4 also shows that Transformers perform worse on *Indirect object in PP datives* (20c) compared to *Passive indirect objects* (20b), although neither subcase introduces long predicate-argument dependencies.

(19) PP within subject NPs:
[**A cat** on a mat]$_{\textbf{subj}}$ **ate** a fish.

(20) Sub-cases of PP within indirect object NPs:

| Generalization cases | Vanilla Transformer | | T5 | | LLaMa | | AM-Parser |
|---|---|---|---|---|---|---|---|
| Deeper PP recursion | $13.1_{\pm1.5}$ | $13.1_{\pm1.5}$ | $15.7_{\pm0.7}$ | $16.6_{\pm1.0}$ | $19.8_{\pm1.1}$ | $20.6_{\pm1.0}$ | $100.0_{\pm0.0}$ |
| Deeper tail CP recursion | $0.2_{\pm0.1}$ | $0.9_{\pm0.3}$ | $0.8_{\pm0.2}$ | $5.3_{\pm0.4}$ | $3.9_{\pm0.4}$ | $12.1_{\pm0.7}$ | $100.0_{\pm0.0}$ |
| Deeper center embedding | $0.0_{\pm0.0}$ | $0.0_{\pm0.0}$ | $0.0_{\pm0.0}$ | $0.0_{\pm0.0}$ | $0.0_{\pm0.0}$ | $0.0_{\pm0.0}$ | $99.5_{\pm0.4}$ |
| Shallower PP recursion | $98.7_{\pm0.8}$ | $98.7_{\pm0.8}$ | $90.2_{\pm2.2}$ | $93.1_{\pm1.9}$ | $97.3_{\pm0.9}$ | $98.9_{\pm0.6}$ | $100.0_{\pm0.0}$ |
| Shallower tail CP recursion | $32.6_{\pm3.6}$ | $55.2_{\pm4.2}$ | $44.8_{\pm2.8}$ | $60.9_{\pm2.1}$ | $85.4_{\pm3.6}$ | $98.1_{\pm0.7}$ | $100.0_{\pm0.0}$ |
| Shallower center embedding | $0.0_{\pm0.0}$ | $0.0_{\pm0.0}$ | $0.0_{\pm0.0}$ | $64.1_{\pm19.1}$ | $0.0_{\pm0.0}$ | $50.7_{\pm5.7}$ | $100.0_{\pm0.0}$ |
| PP in subject NPs | $0.0_{\pm0.0}$ | $0.0_{\pm0.0}$ | $0.0_{\pm0.0}$ | $0.8_{\pm0.5}$ | $12.3_{\pm4.4}$ | $28.9_{\pm3.5}$ | $57.6_{\pm8.1}$ |
| PP in indirect object NPs | $42.5_{\pm2.2}$ | $42.5_{\pm2.2}$ | $50.1_{\pm1.7}$ | $53.8_{\pm1.4}$ | $55.0_{\pm3.9}$ | $71.2_{\pm4.2}$ | $90.4_{\pm8.1}$ |
| RC in subject NPs | $0.0_{\pm0.0}$ | $0.0_{\pm0.0}$ | $0.0_{\pm0.0}$ | $0.2_{\pm0.2}$ | $3.4_{\pm1.6}$ | $29.5_{\pm3.4}$ | $55.8_{\pm8.4}$ |
| RC in indirect object NPs | $34.4_{\pm6.0}$ | $34.8_{\pm6.1}$ | $35.1_{\pm1.9}$ | $36.6_{\pm2.1}$ | $48.6_{\pm1.9}$ | $55.0_{\pm2.1}$ | $74.4_{\pm6.4}$ |
| Indirect object-extracted RC | $4.7_{\pm5.6}$ | $4.7_{\pm5.7}$ | $0.0_{\pm0.0}$ | $0.0_{\pm0.0}$ | $0.1_{\pm0.3}$ | $2.5_{\pm3.2}$ | $0.0_{\pm0.0}$ |
| Indirect object *wh*-questions | $35.9_{\pm8.3}$ | $42.4_{\pm13.5}$ | $0.0_{\pm0.0}$ | $0.4_{\pm0.7}$ | $27.9_{\pm9.3}$ | $73.5_{\pm18.4}$ | $41.4_{\pm42.4}$ |
| Active subject *wh*-questions | $96.7_{\pm2.6}$ | $97.1_{\pm2.4}$ | $90.5_{\pm4.0}$ | $98.1_{\pm1.7}$ | $92.8_{\pm6.4}$ | $93.3_{\pm6.0}$ | $99.8_{\pm0.6}$ |
| Passive subject *wh*-questions | $27.4_{\pm1.7}$ | $31.9_{\pm5.4}$ | $20.3_{\pm3.8}$ | $100.0_{\pm0.0}$ | $4.8_{\pm4.5}$ | $15.3_{\pm17.5}$ | $100.0_{\pm0.1}$ |
| Direct object *wh*-questions | $2.8_{\pm3.4}$ | $16.0_{\pm12}$ | $47.2_{\pm1.0}$ | $98.5_{\pm0.9}$ | $0.5_{\pm0.5}$ | $8.6_{\pm5.7}$ | $29.4_{\pm33.5}$ |
| *Wh*-questions with modified NPs | $17.6_{\pm0.9}$ | $17.8_{\pm1.3}$ | $20.5_{\pm1.0}$ | $36.8_{\pm0.4}$ | $15.8_{\pm0.6}$ | $20.8_{\pm2.4}$ | $55.6_{\pm12.5}$ |
| *Wh*-questions long movement | $4.0_{\pm7.8}$ | $4.9_{\pm9.5}$ | $23.3_{\pm4.3}$ | $24.9_{\pm5.1}$ | $0.8_{\pm1.4}$ | $3.0_{\pm4.7}$ | $0.0_{\pm0.0}$ |
| **Overall** | $24.2_{\pm1.0}$ | $27.1_{\pm2.0}$ | $23.4_{\pm1.1}$ | $\mathbf{40.6}_{\pm1.0}$ | $27.6_{\pm1.0}$ | $\mathbf{40.1}_{\pm1.8}$ | $\mathbf{70.8}_{\pm4.3}$ |

Table 5: Mean accuracy (%) using exact-match is shown in gray, accuracy using reformatted exact-match described in Section 4 is shown in black. AM-Parser's graph-based output yields identical scores for both metrics hence only a single column is reported.

a. *Indirect object in double object datives*: Emma **gave** [ a cat on the mat ]$_{iobj}$ **a fish**.

b. *Passive indirect objects*: **A fish was given** to [ a cat on the mat ]$_{iobj}$.

c. *Indirect object in PP datives*: Emma **gave a fish** to [ a cat on the mat ]$_{iobj}$.

The predominant error pattern in the former sub-case was incorrect attachment of PP modifiers to the direct object NP. For example (21b), modifier *the mat* denoted by $x_9$ was attached to *a fish* instead of *the cat*. This suggests that Transformers additionally fall prey to inference patterns akin to a modification rule "attach PPs to NPs in immediate post-verb position", which is compatible with the training data but does not lead to correct generalization.

(21) Gold LF and model-predicted LF for *Emma gave a fish to the cat on the mat*:

    a. Gold: `*cat` $(x_6)$; `*mat`$(x_9)$;
`give.agent` $(x_1,$Emma$)$
$\wedge$ `give.theme` $(x_4, x_3)$ $\wedge$
`give.recipient` $(x_1, x_6)\wedge$
`fish`$(x_3)$ $\wedge$ **`cat`**`.nmod.on` $(x_6, x_9)$

    b. Out: `*cat` $(x_6)$; `*mat`$(x_9)$;
`give.agent` $(x_1,$Emma$)$
$\wedge$ `give.theme` $(x_4, x_3)$ $\wedge$
`give.recipient` $(x_1, x_6)\wedge$
`fish`$(x_3)$ $\wedge$ **`fish`**`.nmod.on` $(x_3, x_9)$

## C.3 RC Modifiers in unseen positions

Generalizing RC modifiers to unseen positions presents a similar challenge as PP modification cases, due to unobserved long-distance dependencies. As shown in Table 6, all models exhibit a significant performance discrepancy between constructions involving unseen long predicate-argument dependencies and those that do not.

For novel positions that introduce long predicate-argument dependencies, RC modification in the indirect object appears more difficult than in the subject position, contrary to the case with PP modifiers. The primary error pattern (22) demonstrates that models struggle to detect the RC boundary when the relative clause ends with a verb. They systematically misinterpret the indirect object `a fish` of the main verb `gave` as the direct object of the adjacent embedded verb `slept`.

(22) Gold LF and model-predicted LF for *Emma gave a cat that slept a fish*:

    a. Gold: `give.agent` $(x_1,$Emma$)$
$\wedge$ `give.`**`recipient`** $(x_1, x_3)$
$\wedge$ `give.theme` $(x_1, x_7)\wedge$
`cat`$(x_3)$ $\wedge$ `cat.nmod` $(x_3, x_5)$ $\wedge$
`sleep.agent`$(x_5, x_3)$ $\wedge$ `fish`$(x_7)$

    b. Out: `give.agent` $(x_1,$Emma$)$
$\wedge$ `give.theme` $(x_1, x_3)$ $\wedge$
`cat`$(x_3)$ $\wedge$ `cat.nmod` $(x_3, x_5)$
`sleep.agent`$(x_5, x_3)$ $\wedge$
`sleep.theme`$(x_5, x_7)$ $\wedge$`fish`$(x_7)$

| Generalization cases | Long pred-arg dependency? | Vanilla Transformer | T5 | LLaMa | AM parser |
|---|---|---|---|---|---|
| Sub-case: Passive indirect objects
**A fish was given** to [ a cat that slept ]$_{\text{iobj}}$. | ✗ | $72.0_{\pm 6.6}$ | $74.2_{\pm 2.7}$ | $97.1_{\pm 1.2}$ | $99.5_{\pm 0.6}$ |
| Sub-case: Indirect object in PP datives
Emma **gave a fish** to [ a cat that slept ]$_{\text{iobj}}$. | ✗ | $27.0_{\pm 9.8}$ | $38.9_{\pm 5.3}$ | $72.7_{\pm 7.8}$ | $99.3_{\pm 1.1}$ |
| Sub-case: Indirect object in double object datives
Emma **gave** [ a cat that slept ]$_{\text{iobj}}$ **a fish**. | ✓ | $7.9_{\pm 8.5}$ | $0.2_{\pm 0.2}$ | $0.3_{\pm 0.3}$ | $28.9_{\pm 17.2}$ |
| Subject
[**A cat** that slept]$_{\text{subj}}$ **ate** a fish. | ✓ | $0.0_{\pm 0}$ | $0.2_{\pm 0.2}$ | $29.4_{\pm 3.4}$ | $51.7_{\pm 8.4}$ |

Table 6: Performance of RC modification generalization broken down by construction.

## C.4 Gap constructions

While performing poorly on indirect object-extracted relative clauses (23), all tested models systematically mirror the direct object-extracted RC pattern in training, as demonstrated by the incorrect output (23b). They furthermore show distinct difficulties when handling *wh*-questions cases, as will be discussed in the remainder of this section.

(23) Input: Ella cooked the servant that Emma gave a tool to __ .

   a. Gold: `*servant`$(x_3)$`;cook.agent`$(x_1,$ `Ella`$) \land$ `cook.theme`$(x_1, x_3)$ $\land$ `servant.nmod(` $x_3, x_6)$ $\land$ `give.agent`$(x_6,$ `Emma`$)$ $\land$ `give.theme` $(x_6, x_8)$ $\land$ `give.recipient`$(x_6, x_3) \land$ `tool` $(x_8)$

   b. Models output: `*servant`$(x_3)$`;cook.agent`$(x_1,$ `Ella`$) \land$ `cook.theme`$(x_1, x_3)$ $\land$ `servant.nmod(` $x_3, x_6)$ $\land$ `give.agent`$(x_6,$ `Emma`$)$ $\land$ `give.theme` $(x_6, x_3) \land$ `give.recipient`$(x_6, x_8) \land$ `tool` $(x_8)$

### C.4.1 Direct and indirect *wh*-questions

The Transformer trained from scratch and LLaMa frequently misinterpret the theme role in direct object *wh*-questions. For example, they often fail to map *wh*-words to '?' as illustrated in (24b):

(24) Input: What did Emma sell to Liam ?

   a. Gold:`sell.theme` $(x_3,$ `?`$) \land$ `sell.agent` $(x_3,$ `Emma`$) \land$ `sell.recipient`$(x_3,$`Liam`$)$

   b. Output of Vanilla Transformer and LLaMa: `sell.theme` $(x_3,$ $x_5) \land$ `sell.agent` $(x_3,$ `Emma`$) \land$ `sell.recipient`$(x_3,$`Liam`$)$

   c. AM-Parser's output: `sell.agent` $(x_3,$ `?`$) \land$ `sell.theme` $(x_3,$ `Emma`$) \land$ `sell.recipient`$(x_3,$`Liam`$)$

This error pattern can be traced back to frequency of the subsequences in the training data. Three

types of tokens can appear post-comma in the output LF space: $x$, `?` denoting *wh*-words, or a proper noun (`PropN`), such as `Emma`. The subsequence `theme`$(x_i, x_j)$ is 20 times more frequent than `theme`$(x_i,$`?`$)$ and `theme`$(x_i,$`PropN`$)$. This discrepancy does not affect all models equally; in fact, T5 can generalize correctly for some constructions despite this skewed label distribution, achieving near-perfect accuracy for direct object *wh*-questions. However, when it comes to less frequent constructions—indirect object *wh*-questions, T5 overgeneralizes. In 94.6% of these cases, it erroneously produces the observed direct object *wh*-questions pattern `theme`$(x_i,$`?`$)$, instead of the correct but unseen `recipient`$(x_i,$`?`$)$. This observation aligns with the findings of Wu et al. (2023); Yao and Koller (2022), who noted that the decoder of Transformer models tends to exhibit a heavy bias towards generating observed $n$-grams.

### C.4.2 *Wh*-questions with long-distance movement

All models achieve very low accuracy when generalizing to longer filler-gap dependency across CPs, but an error analysis shows that Transformer and structure-aware models face distinct challenges. As shown in (25b), the Transformer trained from scratch commonly misinterprets the complementizer *that* (corresponding to `ccomp` in LF) as a relative pronoun (`nmod`). Additionally, it tends to interpret the *wh*-word as the direct object of the CP verb, *e.g., say*. In the most common errors for T5 and LLaMa (25c), the whole gap conjunct (`paint.theme`$(x_7,$`?`$)$) is missing, revealing their difficulties in establishing long-range filler-gap dependencies between the initial *wh*-word and the embedded gap position. On the other hand, AM-Parser cannot decode non-projective dependencies, thus has 0% accuracy (see more detailed discussion of the issue in §D).

(25) Input: What did Liam say that the bear painted __ ?

    a. Gold: `*bear(x₆); say.agent (x₃,Liam) ∧ say.ccomp (x₃,x₇) ∧ paint.agent (x₇,x₆) ∧ paint.theme (x₇,?)`

    b. Output of vanilla Transformer: `*bear(x₆); say.agent (x₃,Liam) ∧` `say.theme (x₃,?)` `∧ say.``nmod` `(x₃,x₇) ∧ paint.agent (x₇,x₆) ∧ paint.theme (x₇,?)`

    c. Output of T5 and LLaMa: `*bear(x₅); say.agent (x₃,Liam) ∧ say.ccomp (x₃,x₇) ∧ paint.agent (x₇,x₅)`

### C.4.3  *Wh*-questions with modified NPs

In *wh*-questions with PP and RC modifiers, even though the SLOG training set only contains *wh*-questions with unmodified NPs, all models generalize well (accuracy > 80%) to direct object NPs with modifiers (e.g., *Who ate a cake on the table?*). These are cases where the modification pattern is observed in training as a part of declarative sentences. In contrast, performance declines when models encounter *wh*-questions with modifiers in the indirect object position (i.e., modification structure not observed as part of declarative sentences). Similarly, for *wh*-questions with subject position modifiers, the performance is very low: both T5 and vanilla Transformer achieve near-zero accuracy, and LLaMa achieves around 5%.

This observation mirrors the patterns discussed in §5.2, attributed to difficulties introduced by unseen subject-verb dependencies across PPs or RCs. In contrast, the structure-aware model exhibits significantly better performance in *wh*-question with subject modification.

### C.4.4  Passive subject *wh*-questions

For subject *wh*-questions, which exhibit no gap, T5 and AM-Parser perform near-perfectly on both active and passive subject questions. Transformer trained from scratch and LLaMa also perform well on active subject questions, but achieve much lower performance on passive subject questions. This performance discrepancy is the most evident in sub-cases where passive subjects function as theme (e.g., (26))—the Transformer trained from scratch has near-zero accuracy for these sub-cases, systematically failing to map *wh*-words to '?' as in (26b):

(26) Input: What was eaten by Emma ?

    a. Gold: `eat.theme (x₂, ?) ∧ eat.agent (x₂, Emma)`

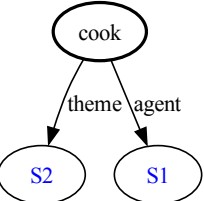

Figure 4: Example of a supertag in an AM dependency tree.

    b. Output of Vanilla Transformer and LLaMa: `eat.theme (x₂,` `x₄``) ∧ eat.agent (x₂, Emma)`

As discussed in Section C.4, this error pattern may result from the highly imbalanced label distribution in training output space. Both LLaMa and Transformer trained from scratch are inclined to repeat the substantially more common subsequence `theme`$(x_i, x_j)$ over `theme`$(x_i, ?)$.

## D  AM-Parser-specific issues

While the AM-Parser achieves strong performance on most generalization types, it faces systematic difficulties in handling novel gap structures. In particular, its accuracy on *wh*-questions involving long-distance movement and indirect object-extracted relative clauses is always 0. Additionally, its accuracy significantly fluctuates across runs for both direct and indirect object *wh*-questions. Here, we give a detailed explanation of error patterns for these challenging types.

**Background**  The AM-Parser maps input sentences to graphs by parsing each input sentence to an *AM dependency tree*, which is then deterministically evaluated to a graph (Groschwitz et al., 2018). In the AM dependency tree, each token is labeled with a *supertag*—a small graph illustrated in Figure 4—that captures the lexical meaning of the token. The tree's edges represent the compositional structure of the sentence, which specifies how the meaning of the sentence is recursively computed from the supertags. The supertag in Figure 4 represents the meaning of *cooked* in the sentence *Ella cooked the servant that Emma gave a tool to*. The blue markers "S1" and "S2" indicate that two arguments are still needed to fill the agent and theme roles of *cook*.

***Wh*-questions with long movement**  We show an example of a predicted AM dependency tree for a *wh-question with long movement* in Figure 5

and the corresponding gold AM dependency tree in Figure 6. As discussed in Section 5.3, the parser used in this paper is limited to predicting projective AM dependency trees, but the gold AM dependency tree in Figure 6 is non-projective (the edge `snapped -> Who` crosses the edge `root -> appreciate`). Thus it is impossible for the AM-Parser to predict the correct compositional structure.

Instead of the A* parser, one could instead use the fixed-tree decoder of Groschwitz et al. (2018), which is capable of predicting non-projective AM dependency trees. This parser achieves nonzero accuracy (36%) on *wh*-questions with long movement, confirming our hypothesis that the projectivity is the issue. However, the A* parser outperforms the fixed-tree decoder on most other generalization types, which is why we only report its results in the main body of the paper. The transition-based AM-Parser of Lindemann et al. (2020) can also predict non-projective trees, but uses a different probability model that is incompatible with the training algorithm of Groschwitz et al. (2021) that we use here.

Note that the A* AM-Parser shares its limitation to projective structures with many other structure-aware models. For instance, the LeAR model of Liu et al. (2021) uses phrase-structure trees as compositional structures, and the CSL-T5 parser of Qiu et al. (2022a) uses phrase-structure trees during the data augmentation process. Because phrase structure trees are equivalent to projective dependency trees, they are likely to encounter similar difficulties on SLOG.

**Direct & indirect *wh*-questions and indirect object-extracted RC** The AM-Parser consistently shows zero accuracy for indirect object-extracted RCs and exhibits big performance fluctuation across different runs for direct and indirect *wh*-questions. This is because in these generalization types, the gold meaning representations in the test set require supertags that are infrequent in training.

We show an example of AM dependency trees for a *direct object wh-question* in Figure 7, with gold supertags in Figure 7a and predicated supertags in Figure 7b. The issue here is that the model predicts the wrong supertag for *sell*, treating *What* as its `agent` instead of `theme`, and *Emma* as its `theme` rather than `agent`, which results in the erroneous output LF as shown in (24c). The

AM-Parser is limited to using supertags that it observed during training (possibly with different node labels to accommodate novel lexical material). For the direct *wh*-question case, the correct supertag was actually present in the training data, but was much less frequent than the erroneous one in Figure 7b. We observe a similar discrepancy in the frequency distribution between predicted and gold supertags for indirect object-extracted RCs and indirect *wh*-questions.

We conjecture that the AM-Parser was overly sensitive to the supertag distribution in the training data, pointing to a further architectural limitation. Thus, while the AM-Parser can compensate the distribution shift of the meaning representations as a whole, SLOG exposes its weakness to distribution shifts in the lexical supertags.

## E   Results with variable-free LFs

Table 7 reports the accuracy on SLOG using variable-free logical forms. The AM-Parser is unable to handle the variable-free format and therefore is omitted. The hyperparameters for the three tested models are the same as the experiments described in Appendix A.

The variable-free LF, as discussed in Appendix B and Wu et al. (2023), exhibits certain limitations and ambiguities which render direct comparisons with variable-based LF results inappropriate. Regardless, all three models achieve higher accuracy scores on the variable-free LFs compared to the COGS LFs, with pretrained models experiencing a particularly significant boost. This aligns with the observations of Qiu et al. 2022b.

Despite the change in LF, the overall trends and challenges remain consistent. The Transformer trained from scratch struggles with the same generalization cases, failing to extrapolate to deeper recursion depths and struggling with cases involving unseen long-distance dependencies. Pretrained models, while exhibiting better overall performance, continue to struggle with more structurally complex generalization cases in their respective categories. These include deeper center embedding, indirect object-extracted RC and *wh*-questions with long movement.

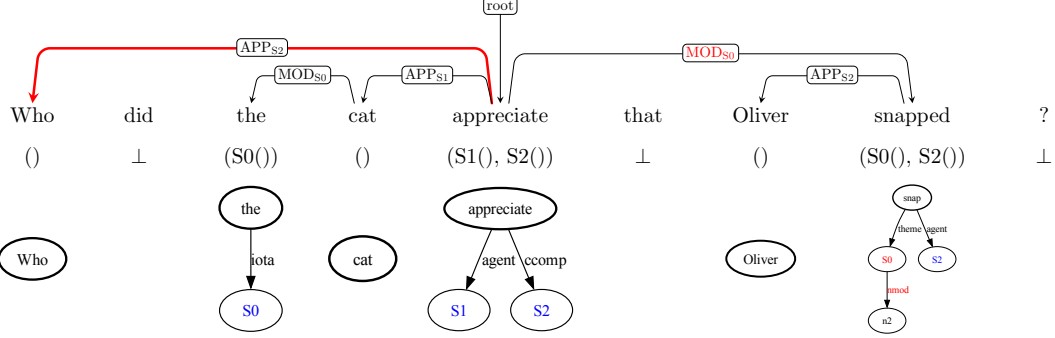

Figure 5: Example of predicted AM dependency tree for *wh*-questions with long movement

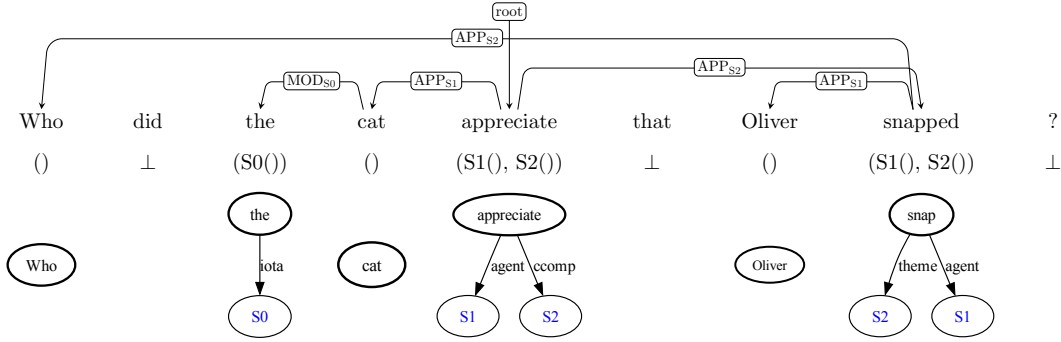

Figure 6: Example of gold AM dependency tree for *wh*-questions with long movement

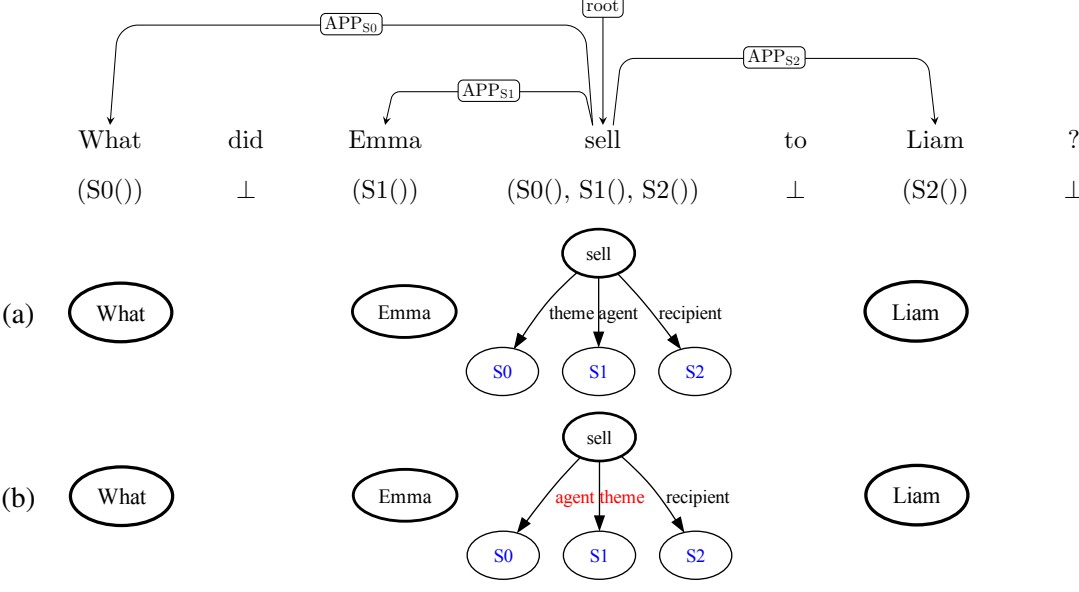

Figure 7: AM dependency tree for a direct object *wh*-question. (a) displays the gold supertags and (b) shows the incorrect predicted supertags.

| Generalization cases | Vanilla Transformer | T5 | LLaMa |
|---|---|---|---|
| Deeper PP recursion | $7.8_{\pm1.8}$ | $63.0_{\pm2.9}$ | $90.9_{\pm3.3}$ |
| Deeper tail CP recursion | $1.0_{\pm0.5}$ | $46.2_{\pm2.6}$ | $44.1_{\pm7.9}$ |
| Deeper center embedding | $0.0_{\pm0.0}$ | $7.8_{\pm1.1}$ | $9.4_{\pm2}$ |
| Shallower PP recursion | $98.2_{\pm1.6}$ | $99.6_{\pm0.9}$ | $100.0_{\pm0.0}$ |
| Shallower tail CP recursion | $89.3_{\pm3.3}$ | $99.3_{\pm1.6}$ | $100.0_{\pm0.0}$ |
| Shallower center embedding | $0.1_{\pm0.2}$ | $99.8_{\pm0.3}$ | $99.8_{\pm0.4}$ |
| PP in subject NPs | $0.2_{\pm0.3}$ | $73.2_{\pm9.0}$ | $93.4_{\pm4.8}$ |
| PP in indirect object NPs | $29.3_{\pm10.7}$ | $97.4_{\pm2.1}$ | $98.1_{\pm1.9}$ |
| RC in subject NPs | $0.1_{\pm0.1}$ | $60.8_{\pm6.3}$ | $73.9_{\pm13.5}$ |
| RC in indirect object NPs | $4.0_{\pm1.9}$ | $71.9_{\pm0.8}$ | $73.6_{\pm3.9}$ |
| Indirect object-extracted RC | $0.0_{\pm0.0}$ | $62.4_{\pm7.5}$ | $3.3_{\pm2.8}$ |
| Indirect object *wh*-questions | $34.1_{\pm31.1}$ | $93.4_{\pm4.8}$ | $83.8_{\pm11.3}$ |
| Active subject *wh*-questions | $99.0_{\pm0.5}$ | $99.8_{\pm0.3}$ | $96.2_{\pm2.6}$ |
| Passive subject *wh*-questions | $57.3_{\pm23.8}$ | $99.9_{\pm0.1}$ | $96.0_{\pm3.0}$ |
| Direct object *wh*-questions | $41.8_{\pm3.8}$ | $48.4_{\pm0.4}$ | $44.1_{\pm4.6}$ |
| *Wh*-questions with modified NPs | $18.1_{\pm2.3}$ | $68.0_{\pm1.9}$ | $69.4_{\pm6.8}$ |
| *Wh*-questions long movement | $7.4_{\pm3.7}$ | $45.6_{\pm4.6}$ | $35.7_{\pm6.5}$ |
| Total | $28.7_{\pm4.1}$ | $72.7_{\pm1.1}$ | $71.3_{\pm3}$ |

Table 7: Mean accuracy (%) on SLOG using the variable-free logical form of Qiu et al. (2022a).