# OpenReview forum: "SLOG: A Structural Generalization Benchmark for Semantic Parsing"
_EMNLP/2023/Conference — EMNLP 2023 Main_

### Official Review · Reviewer_t68a · 2023-08-02

**Soundness:** 4

**Excitement:**

4: Strong: This paper deepens the understanding of some phenomenon or lowers the barriers to an existing research direction.

**Paper Topic And Main Contributions:**

The paper expands the COGS benchmark (Kim and Linzen, 2020) to evaluate semantic parsers on a broader set of compositional generalization phenomena. For instance, if training data contains semantic parses for sentences of the form "Noah saw the cat that froze", i.e. NP VERB NP_WITH_RELATIVE_CLAUSE, then the benchmark tests whether it can generalize to sentences of the form "The cat that froze smiled.", even if it hasn't seen a sentence of the form NP_WITH_RELATIVE_CLAUSE VERB and its corresponding semantic parse. In addition to extending the COGS benchmark, they use their new benchmark (called SLOG) to conduct an empirical investigation of the effectiveness of several semantic parsing approaches: a vanilla transformer, some fine-tuned LLMs, and something called the AM-Parser, based on a model that explicitly incorporates syntactic structure.

**Reasons To Accept:**

The extension of COGS is nontrivial, including 17 new phenomena into the SLOG benchmark. Moreover, they improve the evaluation criteria (which was "exact match" in COGS) to make it more robust by scoring any parse as correct if it is semantically equivalent to the gold parse. They also show that SLOG offers a different perspective than COGS. While COGS scores the fine-tuned LLM approaches similarly to the AM-Parser, SLOG highlights more of the benefits provided by a structure-aware approach like AM-Parser. I suppose one could cynically suggest that the SLOG benchmark was designed to favor structure-aware approaches and highlight their advantages, but regardless it offers an alternative perspective and a finer-grained way to assess the relative strengths and weaknesses of different approaches to semantic parsing.

**Reasons To Reject:**

I wish that there had more justification of *why* the new unseen patterns were deemed to be learnable from the patterns included in the training data. For instance, I guess there are some Occam's razor reasons to see sentences like "What did the cat see?" and "Emma said that the cat saw a fish" and then to infer the meaning of "What did Emma say that the cat found?", but why should such a generalization be preferred over others, in the absence of any examples?

Also, I might have liked more justification of *whether it was important* that the new unseen patterns were learnable. Particularly in terms of the recursive forms, some of which are highly unnatural (who embeds 12 center clauses in practice?) and seem to be asking the semantic parser to OVER-generalize. A sentence like "Ava saw the ball in the bottle on the table." is inherently ambiguous -- are we rewarding systems for jumping to the conclusion of a tail-recursive interpretation when that isn't how human language would operate?



**Reproducibility:**

4: Could mostly reproduce the results, but there may be some variation because of sample variance or minor variations in their interpretation of the protocol or method.

**Reviewer Confidence:**

3: Pretty sure, but there's a chance I missed something. Although I have a good feel for this area in general, I did not carefully check the paper's details, e.g., the math, experimental design, or novelty.

**Typos Grammar Style And Presentation Improvements:**

Section 2.3, example (8):
Liam saw the boy that Emma GAVE a cake to.

---

> ### Author Rebuttal · Authors · 2023-08-28
>
> Thank you for your review!
>
> >  I might have liked more justification of whether it was important that the new unseen patterns were learnable. Particularly in terms of the recursive forms, some of which are highly unnatural (who embeds 12 center clauses in practice?) and seem to be asking the semantic parser to OVER-generalize
>
> The generalization cases in SLOG are largely motivated by the observed frequency gaps in natural language, aiming to generalize from common patterns to their less common counterparts. For instance, typical recursion depths to deep ones, common extraction positions to less common ones. Given the natural data distribution, less frequent cases are more likely to be underrepresented in the training data---like you also point out, large depths of center embeddings are unlikely to occur naturally. However, given such an input (insofar as it is well-formed), the ideal behavior of an NLP model is to still provide an appropriate meaning representation rather than to fail. Therefore, it’s desirable that models can extrapolate from frequent patterns to interpret the rare scenarios. Thus, we believe it's worth assessing models on this capability (regardless of human ability to generalize under the same conditions or human performance limitations).
>
> We refer the reviewer to our response to R1 regarding more detailed rationale behind testing deep recursive patterns that may be unnatural and/or humans may face limitations in processing.
>
>
> > I wish that there had more justification of why the new unseen patterns were deemed to be learnable from the patterns included in the training data. For instance, I guess there are some Occam's razor reasons to see sentences like "What did the cat see?" and "Emma said that the cat saw a fish" and then to infer the meaning of "What did Emma say that the cat found?", but why should such a generalization be preferred over others, in the absence of any examples?
>
> The target generalizations all share the property that they can be compositionally built up from parts that constitute the training examples. Still, like you correctly point out, multiple generalizations may be plausible given the building blocks available in the training data. Nevertheless, arriving at the meaning that is compatible with how humans (i.e., users of the system) would interpret the target example is the ideal, desired behavior of an NLP system when the training data contains distributional gaps (this also relates to our motiviation regarding frequency gaps discussed above). This is equivalent to testing whether our models have the right inductive bias to arrive at the desired interpretation of unobserved complex expressions. We again note that this is a desideratum we set for NLP models (we want our models to arrive at the target meaning given limited data), which may deviate from questions regarding whether human learners can or do generalize this way given the same set of observations.
>
>
> > A sentence like "Ava saw the ball in the bottle on the table." is inherently ambiguous -- are we rewarding systems for jumping to the conclusion of a tail-recursive interpretation when that isn't how human language would operate?
>
> Regarding structural ambiguity in regular English, any corpus with complex sentences will face this issue. Our choices were either to uniformly disambiguate all sentences, which we did, or to force parsers to learn disambiguation, which is potentially diverting from our paper's main focus.
>
>
> We thank the reviewer again for the insightful comments and will integrate these justifications into the revised paper.

---

### Official Review · Reviewer_b6uL · 2023-08-04

**Soundness:** 4

**Excitement:**

4: Strong: This paper deepens the understanding of some phenomenon or lowers the barriers to an existing research direction.

**Paper Topic And Main Contributions:**

This paper provides a dataset (SLOG) aimed at probing the compositional generalization of semantic parsing models at the "structural" level, compared to the mostly "lexical" level that COGS (previous work) is meant to probe for. The authors show that standard transformer baselines struggle deeply on this set. A structure-aware parser that achieves near perfect accuracy on COGS fails to do so on COGS, indicating that this set is better able to test parsers at the upper end of the spectrum (on the axis of compositional generalization).

**Reasons To Accept:**

New datasets are always appreciated. I think this one is thoughtfully constructed, non-trivial, and will be of use to other researchers.

**Reasons To Reject:**

It is probably time to expand these datasets outside of English. Only measuring the performance of one performance-aware parser is a little disappointing. In general, the experiments in this paper are not particularly exciting, though given that it is a dataset paper, I think this is acceptable.

**Reproducibility:**

5: Could easily reproduce the results.

**Reviewer Confidence:**

3: Pretty sure, but there's a chance I missed something. Although I have a good feel for this area in general, I did not carefully check the paper's details, e.g., the math, experimental design, or novelty.

---

> ### Author Rebuttal · Authors · 2023-08-28
>
> Thank you for your review!
>
> > It is probably time to expand these datasets outside of English. Only measuring the performance of one performance-aware parser is a little disappointing.
>
> We agree with the importance of expanding our corpus beyond English and incorporating diverse symbolic parsers. We are considering these aspects as a part of future work. Some of the internal discussions we had regarding these points are:
> * Including Chinese and German given their distinct linguistic properties compared to English. For example, Chinese uses left-branching for modifiers, center-embedding in German comes with gender marking.
> * Adding results from additional structure-aware parsers, for instance, LeAR (Liu et al. 2021) and the decompositional prompting approach of Drozdov et al. (2022).
>
> We thank the reviewer again for their insightful comments, and we would love to include results using other structure-aware parsers to the camera-ready as discussed above, if the paper is accepted.
>
>
> Liu et al. (2021): https://aclanthology.org/2021.findings-acl.97/
>
> Drozdov et al. (2022): https://arxiv.org/abs/2209.15003

---

### Official Review · Reviewer_up6o · 2023-08-05

**Soundness:** 4

**Excitement:**

4: Strong: This paper deepens the understanding of some phenomenon or lowers the barriers to an existing research direction.

**Paper Topic And Main Contributions:**

This paper proses SLOG, a benchmark dataset made for evaluating a model's compositional generalization performance on a set of syntactic patterns in the task of semantic parsing.  The extended syntactic patterns focus on recursion and filler-gap dependencies.  Compared to previous work, recursion depths are deeper and include center-embedding patterns, NPs can have relative clause modifiers, and PP-modified NPs can appear outside of the object position.  The data is produced by synchronous tree grammars which produce both the example and its corresponding logical form.  A wide set of relevant models is evaluated, and existing methods are shown to fail to reach high performance on the task, making the benchmark a useful resource for testing the linguistic understanding of ever more performant models, and where it is difficult to find cases unseen in ever increasing training set sizes.

**Questions For The Authors:**

a.) How do the bounds in the syntactic patterns compare to those in natural language?  For instance, are 4+ depth center-embedding clauses reasonably understandable by humans?  Similarly, it seems the max output token lengths is very long -- maybe beyond what a human could keep track of?  Even for a benchmark with a more theoretical leaning, this is generally good information to present, and is especially pertinent when using pre-trained models as baselines.  This is touched upon a little in the related work section (~504) but is not discussed specifically in the context of the depth and length bounds of this work.


**Reasons To Accept:**

- A well-thoughtout benchmark which extends existing benchmarks in important ways for studying the theoretical linguistic capabilities of recent models.

- Generally well-written paper, lots of good references, clear understanding of the state of the field and related literature.

- A respectable set of baseline models.  Clear demonstration of the inability of existing models to easily solve the more difficult benchmark examples.

**Reasons To Reject:**

- In total, this is only a small extension of the existing benchmark.

- Many of the task examples appear to be of theoretical interest only.

- The actual predictive task is not well described.  Some assumption of familiarity with COGS, its contents and the task definition, seems implicit.

**Reproducibility:**

3: Could reproduce the results with some difficulty. The settings of parameters are underspecified or subjectively determined; the training/evaluation data are not widely available.

**Reviewer Confidence:**

4: Quite sure. I tried to check the important points carefully. It's unlikely, though conceivable, that I missed something that should affect my ratings.

**Typos Grammar Style And Presentation Improvements:**

- Some mild confusion in presentation in that SLOG is presented as an extension of COGS, but this conflicts with the initial motivation of COGS being for lexical generalization and SLOG being for structural generalization.  The language is also midly confusing in other areas -- L069, regarding the AM-Parser performing well on the structural generalization cases in COGS.  It becomes clearer later in the paper that COGS does test structural generalization to a limited depth, but the exposition can come across as contradictory at some points.

- For those who are not familiar with COGS, it could be made clearer what the inputs/outputs are for the prediction task.

- It would have been good to include some structured stack-based transduction models when dealing with modeling recursion.

L314, confusing phrasing

L545, typo, compositional

---

> ### Author Rebuttal · Authors · 2023-08-28
>
> Thank you for your review!
>
> > How do the bounds in the syntactic patterns compare to those in natural language? For instance, are 4+ depth center-embedding clauses reasonably understandable by humans? Similarly, it seems the max output token lengths is very long -- maybe beyond what a human could keep track of?
>
> We thank the reviewer for drawing attention to this important aspect.
>
> In natural language, the depth of embedding is rarely greater than five (Karlsson, 2010). SLOG tests deeper recursive patterns up to depth 12. While this may surpass human processing abilities due to memory constraints, it remains grammatical, echoing Chomsky’s competence versus performance distinction. We also note that our goal with SLOG is to evaluate the linguistic competence  of NLP models, whose goal is not to simulate human performance limitations. That is, it is desirable that these structures are correctly interpreted by NLP models, and therefore the difficulty of human processing should not limit the target capacity of the models we build.
>
> We will incorporate the above discussion into the revised draft, as well as providing explicit details on depth and length bounds.
>
> > For those who are not familiar with COGS, it could be made clearer what the inputs/outputs are for the prediction task. [...] The actual predictive task is not well described. Some assumption of familiarity with COGS, its contents and the task definition, seems implicit.
>
> We will make the specific predictive task of COGS and how it relates to SLOG clearer in the revision.
>
> We thank the reviewer again for their insightful comments, and we will furthermore incorporate your suggestions for stylistic and presentation improvements.

---

### Meta-Review · Area_Chair_3Xfj · 2023-09-18

**Recommendation:** 4

**Metareview:**

This paper presents a benchmark dataset for structural generalization in semantic parsing, as an extension to an earlier dataset focusing on lexical generalization. The benchmark is thoughtfully constructed, non-trivial, and useful for studying the theoretical linguistic capabilities of recent models. The reviewers found the paper also generally well-written.

The reviewers noted a few issues with the paper, most of which concern the presentation and can be addressed in the camera-ready version:
- The similarities and differences between COGS and SLOG should be better highlighted and made accessible for readers not familiar with COGS.
- Some of the included patterns seem at odds with human generalization capabilities (e.g. 4+ depth center-embedding clauses, inherently ambiguous sentences that are uniformly disambiguated). A more thorough discussion and justification of the choices made by the authors would be welcome.
- A reviewer also wished an extension of the dataset to languages other than English. While I second this wish, it cannot be viewed as a flaw of the current paper.

---

### Decision · Program_Chairs · 2023-10-07

**Decision:**

Accept-Main

**Comment:**

This paper presents a benchmark dataset for structural generalization in semantic parsing, as an extension to an earlier dataset focusing on lexical generalization. The benchmark is thoughtfully constructed, non-trivial, and useful for studying the theoretical linguistic capabilities of recent models. The reviewers found the paper also generally well-written.

The reviewers noted a few issues with the paper, most of which concern the presentation and can be addressed in the camera-ready version:
- The similarities and differences between COGS and SLOG should be better highlighted and made accessible for readers not familiar with COGS.
- Some of the included patterns seem at odds with human generalization capabilities (e.g. 4+ depth center-embedding clauses, inherently ambiguous sentences that are uniformly disambiguated). A more thorough discussion and justification of the choices made by the authors would be welcome.
- A reviewer also wished an extension of the dataset to languages other than English. While I second this wish, it cannot be viewed as a flaw of the current paper.